# Effect of Power Ultrasonic on the Expansion of Fiber Strands

**Frederik Wilhelm \*, Sebastian Strauß , Raffael Weigant and Klaus Drechsler**

Fraunhofer Institute for Casting, Composite and Processing Technology IGCV, 86159 Augsburg, Germany;
sebastian.strauss@igcv.fraunhofer.de (S.S.); raffael.weigant@igcv.fraunhofer.de (R.W.);
klaus.drechsler@igcv.fraunhofer.de (K.D.)
**\*** Correspondence: frederik.wilhelm@igcv.fraunhofer.de

**Abstract:** The present study investigates the effect of power ultrasonic on the expansion of fiber strands. A potential application of such expansion is in the production process known as closed injection pultrusion. The fiber strand in the pultrusion injection chamber is in compacted form, and so, any expansion of the fiber strand resulting from power ultrasonic should lead to improved fiber wetting. To investigate this, a wetted fiber strand was clamped on two sides and sonicated in the middle from below. The potential expansion of the fiber strand was visually determined through an observation window. The study concluded that power ultrasonic has a minimal to virtually negligible effect on the expansion of both glass and carbon fiber. The degree of expansion remains within a range of 3% maximum, with a standard deviation in the respective midpoint tests of up to 60% for glass fiber and over 100% for carbon fiber. This shows that the fibers are limited in their freedom of movement, and so no expansion can be achieved using power ultrasonic. A further increase in amplitude does not lead to any further expansion but to the destruction of the fibers.

**Keywords:** power ultrasonic; expansion; carbon fiber; glass fiber; pultrusion; closed-injection pultrusion

## 1. Introduction

The use of pultrusion in component production has attracted great attention in recent years, due to such excellent properties as high strength-to-weight ratio and stiffness, and also due to the high potential for automation. In most cases, the fibers are impregnated in an open impregnation bath. Here, the choice is limited to matrix systems with a long processing time or pot life at room temperature [1].

However, in order to further increase productivity, it is necessary to use highly reactive matrix systems, such as polyurethanes [2,3]. Fiber impregnation then involves the use of a closed injection and impregnation chamber (ii-chamber) rather than an open impregnation bath. A number of different types of ii-chambers are already in use in industry [4–7]. However, the complete and uniform impregnation of a fiber strand with a resin system, particularly in the case of carbon fibers, is still a major challenge with this system [8]. While in an open impregnation bath, the fiber strands can be spread out, thus giving the resin system a short flow path, in an ii-chamber, the fibers have to be impregnated as a compressed strand. The effect of this is to reduce and significantly scatter the mechanical properties of the finished component [9].

One possible way of solving this issue is to integrate power ultrasonic (US) into the ii-chamber. US has many applications, such as mixing, homogenizing and dispersing, and it is already in use in numerous sectors (including the biological and chemical industries) [10–12]. The intended effect of US is to expand the fiber strands through their transient cavitation. Transient cavitation occurs at sound pressures with a power density of 10 W/cm$^2$. The tensile strength of the liquid is thereby exceeded [13], so that cavities/bubbles that are either empty (true cavitation) or filled with gas or water vapor are

formed in the liquid [10,12]. Newton first coined the term 'cavitation' in 1704. It had previously been identified as material spalling in the early years of ship propellers [14–16]. The gas bubbles expand over several oscillation periods until their radius more than doubles within half a period before collapsing within a few milliseconds [17]. The conditions prevailing inside the gas bubbles include speeds of 50 to 150 m/s, temperatures of up to 5000 K, and pressures in the range of $10^9$ to $10^{10}$ Pa [10,13,18,19]. Figure 1 contains a schematic representation of the process.

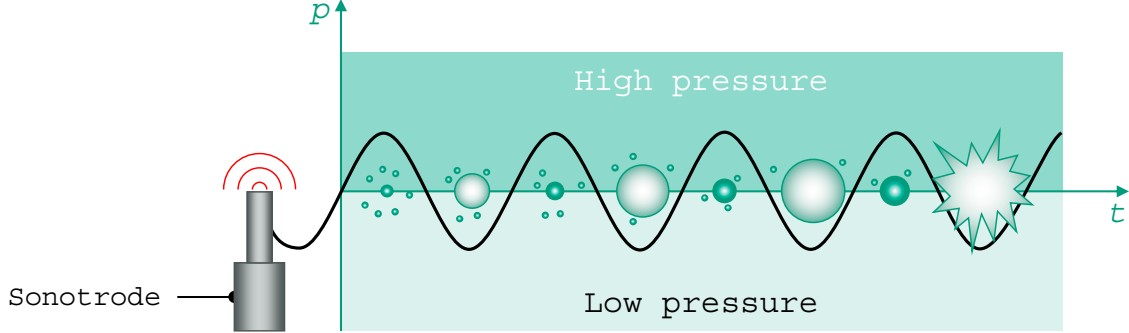

**Figure 1.** Formation of a cavitation bubble during sonication.

The effect of US on composites manufacturing has been previously investigated. These have demonstrated such benefits as enhanced mechanical properties due to better impregnation, improved fiber-matrix adhesion, and increased glass transition temperature [20–23]. Improved component properties were also demonstrated by the use of US in pultrusion [24]. However, few publications have considered the influence of US on fiber distribution. For excitations in the low frequency range, a fiber strand can be compacted further than would be possible using static compaction. Kruckenberg et al. investigated the compacting curve for glass fiber and carbon fiber fabrics at an excitation frequency of 1 to 10 Hz [25]. They found that an increase in fiber volume content of up to 16% is possible for glass fiber fabrics, with up to 6% for carbon fiber fabrics. Gutiérrez et al. conducted similar investigations with comparable results [26]. Glass fiber fabrics were stimulated during compaction and the compaction curve recorded for a frequency range of 10 to 300 Hz and amplitude of 50 to 100 μm. A higher compaction was achieved than with static compaction, especially at low frequencies, accompanied by an increase in fiber volume content of about 10%. The greater compaction was assumed to be due to the more homogeneous distribution of the fibers [27]. The vibration stimulation led to better spreading of the fibers and in turn, to a more uniform fiber distribution [28].

Yamahira et al. showed that US can be employed to attain specific fiber alignments [29]. A beaker was filled with an aqueous sugar solution containing polystyrene fibers, which were then irradiated at frequencies of 25 and 46 kHz. This formed a standing wave in the aqueous sugar solution, and the fibers aligned themselves with this wave.

The aforementioned publications presented the positive effect of the vibration excitation or sonication on the fiber distribution. Investigation of the effect was limited to the compaction and orientation of the fibers. However, they did not consider fiber movement during acoustic irradiation to quantify the possible expansion, which would result in the local increase in permeability referred to at the beginning.

This study therefore focuses on the direct, visual expansion of the fiber strand during sonication, correlated to total area. This enables us to understand the effect of US on fiber strand expansion and, in turn, permeability.

## 2. Materials and Methods

### 2.1. Materials

Preliminary investigations have shown that the Biresin® CR141 epoxy resin system made by Sika, Bad Urach, Germany, which is used as standard in the pultrusion process, becomes cloudy after only a short period of sonication [30]. Any expansion of the fiber strand that might have occurred is then no longer visible.

In order to avoid clouding, a silicone oil (KORASILON® M100 made by Obermeier in Bad Berleburg, Germany) is used instead. It has a viscosity of 100 mPa·s and is thus, comparable to the epoxy resin system. The substitute medium has already been used successfully, as reported in publications in similar fields [1,31].

The type of glass fiber to be used here is SE 3030 with 4800 tex, made by 3B in Herve, Belgium, while the carbon fiber is CT50-4.0/240-E100 with 50k, made by SGL Carbon, Wiesbaden, Germany.

### 2.2. Methods

A test chamber was developed to visually determine the effect of US on fiber distribution and quantify the extent to which US is able to expand a fiber strand. Figure 2 shows the schematic (left) and the real-life assembly (right) of the test chamber. In addition, the schematic representation shows the manipulated variables mass $m$, amplitude $u$ and number of rovings $n$. The expansion $B$ is shown as the test variable. It indicates the extent to which the area of the fiber strand is expanded by US, expressed as a percentage.

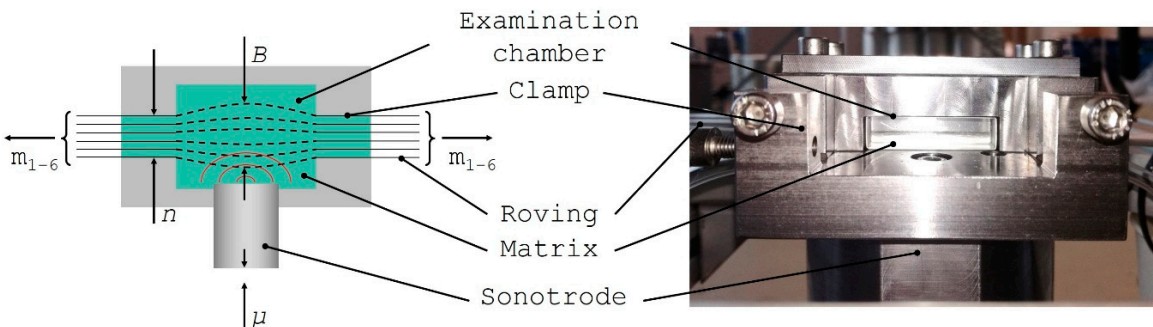

**Figure 2.** Test chamber: schematic representation (**left**) and real-life assembly (**right**, [32]).

Below the test chamber, US is coupled into the examination chamber with a frequency of 20 kHz. The diameter of the sonotrode is 20 mm. The fiber strand is clamped on the left and right and has a fiber volume content of 65% in this area. The fiber volume content is determined by the pultrusion process.

The manipulated variables for the test chamber result from the following parameters:

- Mass per roving $m_n$: The rovings are pulled continuously during pultrusion and are therefore under tension. In the test chamber, the preload on the rovings is simulated by employing an additional weight per roving. This corresponds to the actual stress occurring in the chamber geometries (conical and drop-shaped) [4,6]. The stress is based on the values from the preliminary investigation and varies within the range 250 to 500 g per roving [30].
- Number of rovings $n$: The layer thickness is adjusted by varying the number of rovings. A number of rovings of between 2 and 6 corresponds to a layer thickness of between 2 and 6 mm. The number of rovings determines the average profile thickness in pultrusion.
- Amplitude $u$: The amplitude range can be varied within the range from 12 to 48 μm. Any further increase in amplitude will cause direct damage to the fiber [33].

Silicone oil is injected into the test chamber at a pressure of 1 bar by means of a pneumatic pressure pot. The fiber strand in the test chamber can be viewed through the observation window

(Figure 3). A camera (EOS 500D, Canon, Tokyo, Japan, 15.1 megapixels) records the condition before and during sonication. Afterwards, the percentage changes in the areas with and without sonication are determined and evaluated. One millimeter corresponds to 76 pixels on the digital image.

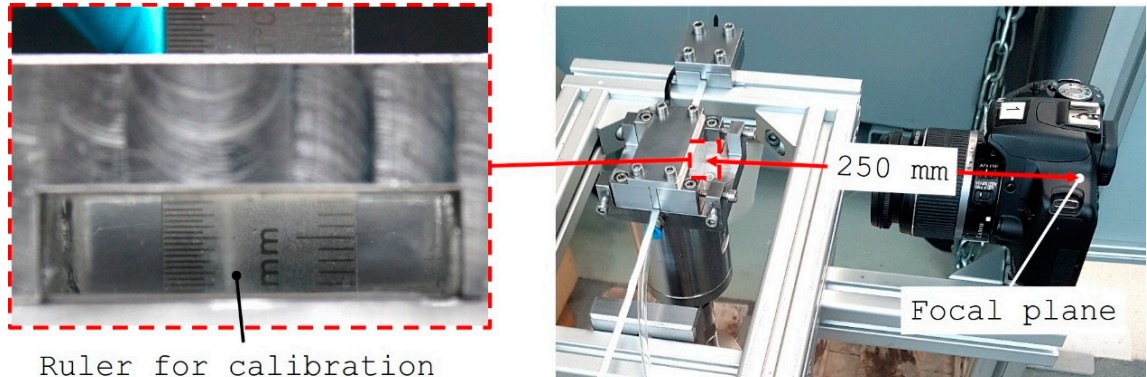

**Figure 3.** Evaluation of expansion *B* using a camera system (**right**), showing the viewfinder image of the observation chamber (**left**), where a calibration ruler can be seen.

The manipulated variables—weight per roving, amplitude, and number of rovings—determine the experiment setup. For the manipulated variables and the glass fiber and carbon fiber materials, the experiment test plan is as shown in Table 1.

**Table 1.** Test plan for the test rig with corresponding manipulated variables.

| Manipulated Variables | Variable | Unit | Level 1 | Level 2 | Midpoint |
|---|---|---|---|---|---|
| Weight per roving | $m$ | g | 250 | 500 | 375 |
| Amplitude | $u$ | μm | 12 | 48 | 30 |
| Number of rovings | $n$ | - | 2 | 6 | 3 |

All tests are performed once, except for the midpoint test, which is repeated three times. This enables a statement to be made about scatter in the experiment setup. All tests are carried out in the sequence given in Table 2.

**Table 2.** Sequence of test steps.

| Section | Activity |
|---|---|
| Setup | Link rovings to weights<br>Insert rovings into test chamber<br>Close test rig<br>Calibrate US |
| Start | Open ball valve of pressure pot<br>Flush test chamber with silicone oil for 30 s at a pressure of 1 bar<br>Start self-timer (5 images after 10 s) |
| $u = 12$ μm | 5 s no US<br>5 s with US<br>5 images at an amplitude of 12 μm; deactivate sonication |
| $u = 48$ μm | 5 s no US<br>5 s with US<br>5 images at an amplitude of 48 μm; deactivate sonication |
| End | Close ball valve of pressure pot |
| Dismantling | Open test chamber and clean |

## 3. Results

The aim of the test chamber is to characterize the effect of US on the expansion of a fiber strand. When it expands, the permeability of the fiber strand increases, thus, facilitating impregnation. The test rig enables explicit investigation of the effect of expansion, by allowing fiber movements to be viewed through an observation window (Figure 3).

### 3.1. Expansion of Glass and Carbon Fiber Strand

Figure 4 shows the test results for glass fiber (left) and carbon fiber (right). The measuring points are as follows: fiber type[G ≜ Glass, C ≜ Carbon]_mass[g]_number of rovings[-]_amplitude[μm].

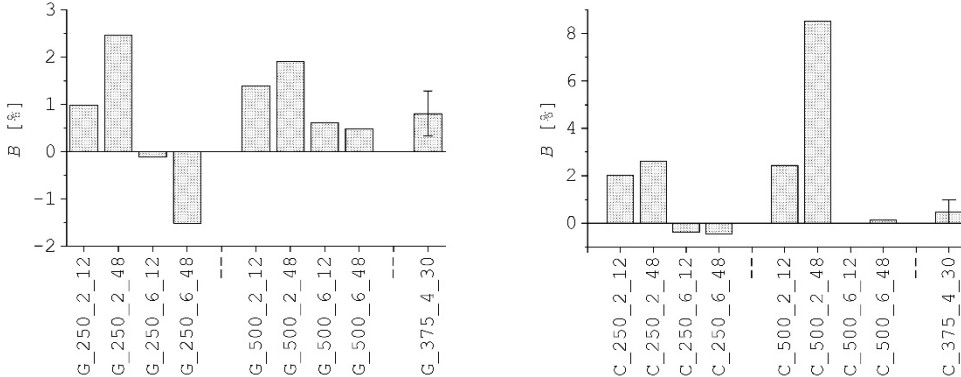

**Figure 4.** Measurements resulting from the investigation of glass fiber expansion (**left**) and carbon fiber expansion (**right**).

The expansion range for glass fiber extends from −1.5% to 2.5% and for carbon fiber from −0.5% to 8.5%. With the exception of an outlier in the carbon fiber test series, the amount of expansion is in the low single-digit range. The midpoint test shows that the measurements can be expected to display a high degree of scatter. A standard deviation of up to 60% for glass fiber and over 100% for carbon fiber were determined in the respective midpoint tests.

### 3.2. Analysing the Effect of Manipulated Variables

In order to determine the effect of the individual manipulated variables, all results were analyzed in effect diagrams. Figure 5 is an example of how effect diagrams (2 and 3) of individual manipulated variables can be formed from a test setup (1) with two manipulated variables (*a* and *b*) and the test variable *y*. The multi-dimensional test setup is reduced to the dimension of the manipulated variable and the mean value per stage determined from the individual measuring points. The effect of a particular manipulated variable is determined by linear interpolation. The effect is deemed positive if the test variable increases as the manipulated variable increases, as shown in the effect diagram (2). Otherwise, the effect is negative (see (3) in Figure 5).

Figure 6 shows the effect of US on the glass fiber for the test variable (expansion B). The effect remains within a range of 3%. Transferred to the reference area without sonication, the area undergoes only minimal change as a result of US. The manipulated variable weight per roving has the greatest positive effect on expansion, followed by amplitude. The effect decreases with increasing layer thickness. The midpoint test shows an average build-up of 0.8%, with a minimum value of 0.5% and a maximum value of 1.4%. In comparison, the effect on expansion for all manipulated variables is largely hidden in the scatter of the test setup.

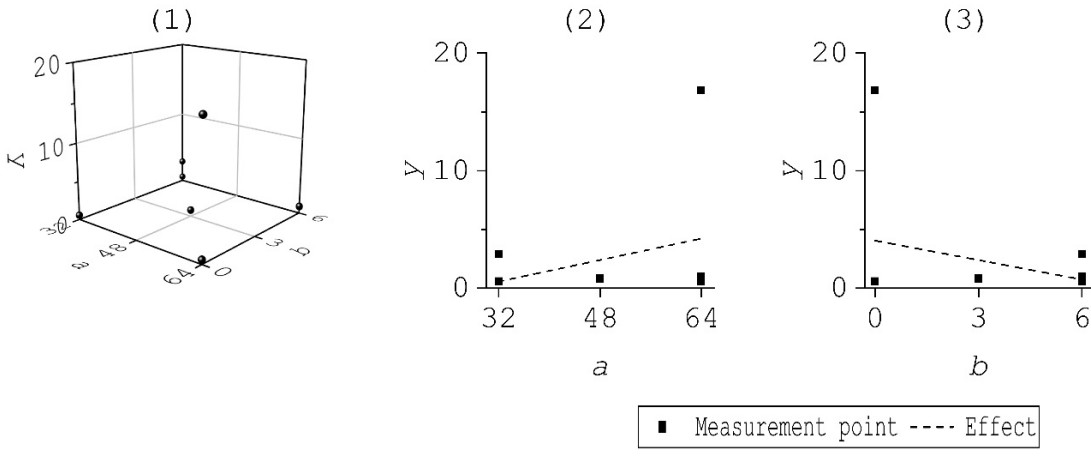

**Figure 5.** Transfer of test space (**1**) to effect diagrams (**2**) and (**3**).

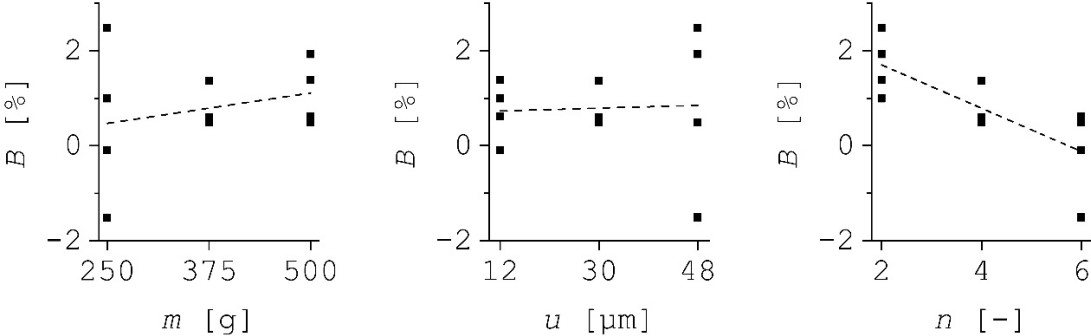

**Figure 6.** Effect diagram for the expansion of glass fiber [30]. Manipulated variables from left to right: mass, amplitude and number of rovings.

A similar situation can be observed in the effect diagram for carbon fiber in Figure 7. The manipulated variable weight per roving has, like the amplitude, the greatest positive effect on expansion. As in the glass fiber tests, the effect on expansion decreases with increasing layer thickness.

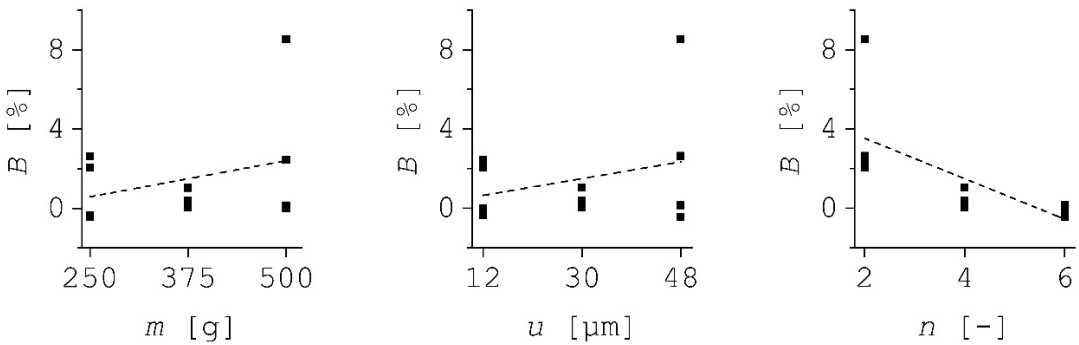

**Figure 7.** Effect diagram for the expansion of carbon fiber [30]. Manipulated variables from left to right: mass, amplitude and number of rovings.

## 4. Discussion

This study investigated the effect of US on the expansion of a fiber strand. The focus of the study was on quantifying the effect during sonication.

The investigation demonstrates that both ultrasonic amplitude and fiber tension have a similarly positive effect on the expansion of a fiber strand. The fact that the expansion increases with increasing fiber tension cannot be explained directly. This is because as the fiber tension increases, any potential

expansion would require correspondingly higher external forces. However, since a relative increase in the fiber strand was indeed determined, the fiber strand is correspondingly more compact at a higher fiber tension. However, US has a small to virtually negligible effect on the expansion of both glass and carbon fiber. The expansion lies within a range of maximum 3%, with a standard deviation of up to 60% for glass fiber and over 100% for carbon fiber, as determined in the midpoint tests. This means that the fibers are limited in their freedom of movement, with the result that no expansion can be achieved by US. Any further increase in amplitude will not lead to further expansion but to destruction of the fibers. The fiber volume content is in direct correlation to the expansion. A minimal reduction in the fiber volume content in the compaction area investigated is equivalent to a minimal increase in permeability. Applied to pultrusion, the mechanical expansion of the fiber strand by US causes only a slight improvement in impregnation. Any improvement to the mechanical properties of the pultruded components as a result of US, as shown in Paper [24], can thus not be attributed to expansion of the fiber strand but to an impact on the resin system.

**Author Contributions:** Conceptualization, F.W.; methodology, F.W., S.S.; validation, F.W., S.S., and R.W.; investigation, S.S., R.W.; data curation, F.W., R.W.; writing—review and editing, F.W., S.S., R.W., K.D.; visualization, F.W., R.W.; project administration, F.W.; funding acquisition, F.W. All authors have read and agreed to the published version of the manuscript.

**Funding:** This research and development project is partly funded by the German Federal Ministry for Economic Affairs and Energy (BMWi) within the Framework Concept "Central Innovation Program for SMEs" and managed by the Project Management Agency AiF Projekt GmbH. The facilities and key technology equipment in Augsburg are funded by the Region of Bavaria; City of Augsburg; BMBF and European Union (in the context of the program "Investing In Your Future"—European Regional Development Fund).

**Conflicts of Interest:** The authors declare no conflict of interest.

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
