# Peer review of "Effect of Power Ultrasonic on the Expansion of Fiber Strands"

_jcs, doi:10.3390/jcs4020050_

Round 1

Reviewer 1 Report

Dear Authors, 

First suggestion, lack of discussion of your tests and phenomena. 

25-26 not in this place the aim of article

63 which mentioned publications???

79- maybe investigated not used

104 what is "ri"

106 documents??? register??

132- Fig. Nomenclature not in the description on Fig. NOt this place.

Best wishes

Reviewer 2 Report

The manuscript is related to the analysis of the effect induced by ultrasonic power for the expansion of fiber strands. A comparison between carbon and glass fibers is reported. 

My recommendation is to not publish the manuscript in the Journal of Composite Science unless some major/minor revisions by the authors.

Major revision:

  • It is not clear in the manuscript is the objective of using ultrasonic power on fiber strands. The fibers can be impregnated of resin, and make larger upon ultrasonic stimulation. Which are the consequences of expanding such fibers in the composite? in terms of mechanical, or other, performance
  • The expansion is in a range of maximum 3 % with a relatively large
    scattering of ±0.5 percentage points." Which is the meaning of the term "scattering"? Please the authors verify the possibility to report such data in terms of normal distribution (average plus/minus standard deviation)
  • Why the amplitude of the ultrasonic power is measured as um? Is it the wavelength?
  • The introduction starts by defining Darcy's law, which is related to the permeability of materials. In the following parts of the manuscript, such a parameter has not been mentioned anymore. The introduction must be re-focused at the beginning on other points, for example on the meaning of modifying fiber strand dimensions in the composites, the current methods adopted, and possible advantages in using ultrasound and ultrasonic power. At this purpose, the authors could refer to the following references:

- Qiao, Jing, Yingrui Li, and Longqiu Li. "Ultrasound-assisted 3D printing of continuous fiber-reinforced thermoplastic (FRTP) composites." Additive Manufacturing 30 (2019): 100926.

- Vannozzi, Lorenzo, et al. "Nanostructured ultra-thin patches for ultrasound-modulated delivery of anti-restenotic drug." International journal of nanomedicine 11 (2016): 69.

- Wang, Junting, et al. "Ultrasound assisted synthesis of Bi2NbO5F/rectorite composite and its photocatalytic mechanism insights." Ultrasonics sonochemistry 48 (2018): 404-411.

  • According to table 2, the test rig has been subjected to two test sessions (12 and 48 um)? If yes, the results obtained at 48 um could be influenced by those achieved at 12 um.
  • "The aim of the test rig is to characterize the effect of the power ultrasonic on expansion of a fiber strand. An expansion increases the permeability of the fiber strand and thus facilitates impregnation." How the authors are sure about the increase of permeability of the fiber if the dimensions change? Could the resin be more compacted instead of penetrating into the fibers?
  • Looking at Figure 6, is the ultrasonic power transmitted to all rigs homogenously, or could it change from rig 1 to rig 6?
  • English grammar must be revised. There are some unclear points in the text

Minor revisions:

  • Ultrasonic con be defined as US
  • The parameter u is not clear. Is it referred to as the amplitude of the frequency of the ultrasound stimulation?
  • Figure 3 (left) is not clear. Which is the meaning of such a depiction?
  • Figure 4. the labels on the X-axis are unclear

Round 2

Reviewer 1 Report

Thank you for improvement.

Reviewer 2 Report

The authors replied to all my revisions